# STRUCTURED PREDICTION AS TRANSLATION BETWEEN AUGMENTED NATURAL LANGUAGES

**Giovanni Paolini, Ben Athiwaratkun, Jason Krone, Jie Ma, Alessandro Achille, Rishita Anubhai, Cicero Nogueira dos Santos, Bing Xiang, Stefano Soatto**

Amazon Web Services

{paoling,benathi,kronej,jieman,aachille,ranubhai,cicnog, bxiang,soattos}@amazon.com

## ABSTRACT

We propose a new framework, *Translation between Augmented Natural Languages* (TANL), to solve many structured prediction language tasks including joint entity and relation extraction, nested named entity recognition, relation classification, semantic role labeling, event extraction, coreference resolution, and dialogue state tracking. Instead of tackling the problem by training task-specific discriminative classifiers, we frame it as a translation task between *augmented natural languages*, from which the task-relevant information can be easily extracted. Our approach can match or outperform task-specific models on all tasks, and in particular, achieves new state-of-the-art results on joint entity and relation extraction (CoNLL04, ADE, NYT, and ACE2005 datasets), relation classification (FewRel and TACRED), and semantic role labeling (CoNLL-2005 and CoNLL-2012). We accomplish this while using the same architecture and hyperparameters for all tasks and even when training a single model to solve all tasks at the same time (multi-task learning). Finally, we show that our framework can also significantly improve the performance in a low-resource regime, thanks to better use of label semantics.

## 1 INTRODUCTION

Structured prediction refers to inference tasks where the output space consists of structured objects, for instance graphs representing entities and relations between them. In the context of natural language processing (NLP), structured prediction covers a wide range of problems such as entity and relation extraction, semantic role labeling, and coreference resolution. For example, given the input sentence *"Tolkien's epic novel The Lord of the Rings was published in 1954-1955, years after the book was completed"* we might seek to extract the following graphs (respectively in a joint entity and relation extraction, and a coreference resolution task):

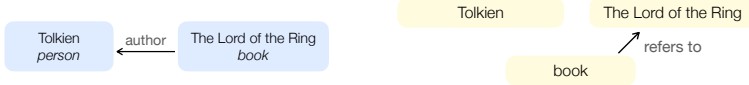

Most approaches handle structured prediction by employing task-specific discriminators for the various types of relations or attributes, on top of pretrained transformer encoders such as BERT (Devlin et al., 2019). Yet, this presents two limitations. First, a discriminative classifier cannot easily leverage latent knowledge that the pretrained model may already have about the meaning (*semantics*) of task labels such as *person* and *author*. For instance, knowing that a *person* can write a *book* would greatly simplify learning the *author* relation in the example above. However, discriminative models are usually trained without knowledge of the label semantics (their targets are class numbers), thus preventing such positive transfer. Second, since the architecture of a discriminative model is adapted to the specific task, it is difficult to train a single model to solve many tasks, or to fine-tune a model from a task to another (*transfer learning*) without changing the task-specific components of the discriminator. Hence, our main question is: can we design a framework to solve different

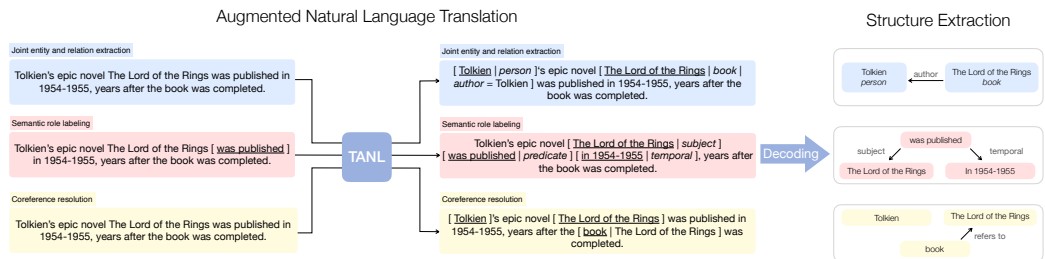

Figure 1: Our TANL model translates between input and output text in *augmented natural language*, and the output is then decoded into structured objects.

structured prediction tasks with the same architecture, while leveraging any latent knowledge that the pre-trained model may have about the label semantics?

In this paper, we propose to solve this problem with a text-to-text model, by framing it as a task of *Translation between Augmented Natural Languages* (TANL). Figure 1 shows how the previous example is handled within our framework, in the case of three different structured prediction tasks. The augmented languages are designed in a way that makes it easy to *encode* structured information (such as relevant entities) in the input, and to *decode* the output text into structured information.

We show that out-of-the-box transformer models can easily learn this augmented language translation task. In fact, we successfully apply our framework to a wide range of structured prediction problems, obtaining new state-of-the-art results on many datasets, and highly competitive results on all other datasets. We achieve this by using the same architecture and hyperparameters on all tasks, the only difference among tasks being the augmented natural language formats. This is in contrast with previous approaches that use task-specific discriminative models. The choice of the input and output format is crucial: by using annotations in a format that is as close as possible to natural language, we allow transfer of latent knowledge that the pre-trained model has about the task, improving performance especially in a low-data regime. Nested entities and an arbitrary number of relations are neatly handled by our models, while being typical sources of complications for previous approaches. We implement an alignment algorithm to robustly match the structural information extracted from the output sentence with the corresponding tokens in the input sentence.

We also leverage our framework to train a single model to solve all tasks at the same time, and show that it achieves comparable or better results with respect to training separately on each task. To the best of our knowledge, this is the first model to handle such a variety of structured prediction tasks without any additional task-specific modules.

To summarize, our key contributions are the following.

1. We introduce TANL, a framework to solve several structure prediction tasks in a unified way, with a common architecture and without the need for task-specific modules. We cast structured prediction tasks as translation tasks, by designing augmented natural languages that allow us to encode structured information as part of the input or output. Robust alignment ensures that extracted structure is matched with the correct parts of the original sentence (Section 3).

2. We apply our framework to (1) joint entity and relation extraction; (2) named entity recognition; (3) relation classification; (4) semantic role labeling; (5) coreference resolution; (6) event extraction; (7) dialogue state tracking (Sections 4 and 5). In all cases we achieve at least comparable results to the current state-of-the-art, and we achieve new state-of-the-art performance on joint entity and relation extraction (CoNLL04, ADE, NYT, and ACE2005 datasets), relation classification (FewRel and TACRED), and semantic role labeling (CoNLL-2005 and CoNLL-2012).

3. We also train a single model simultaneously on all tasks (multi-task learning), obtaining comparable or better results as compared with single-task models (Section 5.1).

4. We show that, thanks to the improved transfer of knowledge about label semantics, we can significantly improve the performance in the few-shot regime over previous approaches (Section 5.2).

5. We show that, while our model is purely generative (it outputs a sentence, not class labels), it can be evaluated discriminatively by using the output token likelihood as a proxy for the class score, resulting in more accurate predictions (Section 3 and Appendix A.3).

The code is available at `https://github.com/amazon-research/tanl`.

## 2 RELATED WORK

Many classical methods for structured prediction (SP) in NLP are generalizations of traditional classification algorithms and include, among others, Conditional Random Fields (Lafferty et al., 2001), Structured Perceptron (Collins, 2002), and Structured Support Vector Machines (Tsochantaridis et al., 2004). More recently, multiple efforts to integrate SP into deep learning methods have been proposed. Common approaches include placing an SP layer as the final layer of a neural net (Collobert et al., 2011) and incorporating SP directly into DL models (Dyer et al., 2015).

Current state-of-the-art approaches for SP in NLP train a task-specific classifier on top of the features learned by a pre-trained language model, such as BERT (Devlin et al., 2019). In this line of work, BERT MRC (Li et al., 2019a) performs NER using two classification modules to predict respectively the first and the last tokens corresponding to an entity for a given input sentence. For joint entity and relation extraction, SpERT (Eberts & Ulges, 2019) uses a similar approach to detect token spans corresponding to entities, followed by a relation classification module. In the case of coreference resolution, many approaches employ a higher-order coreference model (Lee et al., 2018) which learns a probability distribution over all possible antecedent entity token spans.

Also related to this work are papers on sequence-to-sequence (seq2seq) models for multi-task learning and SP. Raffel et al. (2019) describe a framework to cast problems such as translation and summarization as text-to-text tasks in natural language, leveraging the transfer learning power of a transformer-based language model. Other sequence-to-sequence approaches solve specific structured prediction tasks by generating the desired output directly: see for example WDec (Nayak & Ng, 2020) for entity and relation extraction, and SimpleTOD (Hosseini-Asl et al., 2020) and SOLOIST (Peng et al., 2020) for dialogue state tracking. Closer to us, GSL (Athiwaratkun et al., 2020), which introduced the term *augmented natural language*, showed early applications of the generative approach in sequence labeling tasks such as slot labeling, intent classification, and named entity recognition without nested entities. Our approach is also related to previous works that use seq2seq approaches to perform parsing (Vinyals et al., 2015; Dyer et al., 2016; Choe & Charniak, 2016; Rongali et al., 2020), with the main difference that we propose a general framework that uses augmented natural languages as a way to unify multiple tasks and exploit label semantics. In some cases (e.g., relation classification), our output format resembles that of a question answering task (McCann et al., 2018). This paradigm has recently proved to be effective for some structured prediction tasks, such as entity and relation extraction and coreference resolution (Li et al., 2019c; Zhao et al., 2020; Wu et al., 2020). Additional task-specific prior work is discussed in Appendix A.

Finally, TANL enables easy multi-task structured prediction (Section 5.1). Recent work has highlighted benefits of multi-task learning (Changpinyo et al., 2018) and transfer learning (Vu et al., 2020) in NLP, especially in low-resource scenarios.

## 3 METHOD

We frame structured prediction tasks as text-to-text translation problems. Input and output follow specific augmented natural languages that are appropriate for a given task, as shown in Figure 1. In this section, we describe the format design concept and the decoding procedure we use for inference.

**Augmented natural languages.** We use the joint entity and relation extraction task as our guiding example for augmented natural language formats. Given a sentence, this task aims to extract a set of *entities* (one or more consecutive tokens) and a set of *relations* between pairs of entities. Each predicted entity and relation has to be assigned to an entity or a relation type. In all the datasets considered, the relations are asymmetric; *i.e.*, it is important which entity comes first in the relation (the *head* entity) and which comes second (the *tail* entity). Below is the augmented natural language designed for this task (also shown in Figure 1):

> **Input:** Tolkien's epic novel The Lord of the Rings was published in 1954-1955, years after the book was completed.
>
> **Output:** [ Tolkien | *person* ]'s epic novel [ The Lord of the Rings | *book* | *author* = Tolkien ] was published in 1954-1955, years after the book was completed.

Specifically, the desired output replicates the input sentence and augments it with patterns that can be decoded into structured objects. For this task, each group consisting of an entity and possibly some relations is enclosed by the special tokens [ ]. A sequence of |-separated tags describes the entity type and a list of relations in the format "X = Y", where X is the relation type, and Y is another entity (the *tail* of the relation). Note that the objects of interest are all within the enclosed patterns "[ ... | ... ]". However, we replicate all words in the input sentence, as it helps reduce ambiguity when the sentence contains more than one occurrence of the same entity. It also improves learning, as shown by our ablation studies (Section 5.3 and Appendix B). In the target output sentence, entity and relation types are described in natural words (e.g. *person*, *location*) — not abbreviations such as *PER*, *LOC* — to take full advantage of the latent knowledge that a pre-trained model has about those words.

For certain tasks, additional information can be provided as part of the input, such as the span of relevant entities in semantic role labeling or coreference resolution (see Figure 1). We detail the input/output formats for all structured prediction tasks in Section 4.

**Nested entities and multiple relations.** Nested patterns allow us to represent hierarchies of entities. In the following example from the ADE dataset, the entity "lithium toxicity" is of type *disease*, and has a sub-entity "lithium" of type *drug*. The entity "lithium toxicity" is involved in multiple relations: one of type *effect* with the entity "acyclovir", and another of type *effect* with the entity "lithium". In general, the relations in the output can occur in any order.

> **Input:** Six days after starting acyclovir she exhibited signs of lithium toxicity.
>
> **Output:** Six days after starting [ acyclovir | *drug* ] she exhibited signs of [ [ lithium | *drug* ] toxicity | *disease* | *effect* = acyclovir | *effect* = lithium ].

**Decoding structured objects.** Once the model generates an output sentence in an augmented natural language format, we decode the sentence to obtain the predicted structured objects, as follows.

1. We remove all special tokens and extract entity types and relations, to produce a cleaned output. If part of the generated sentence has an invalid format, that part is discarded.
2. We match the input sentence and the cleaned output sentence at the token levels using the dynamic programming (DP) based Needleman-Wunsch alignment algorithm (Needleman & Wunsch, 1970). We then use this alignment to identify the tokens corresponding to entities in the original input sentence. This process improves the robustness against potentially imperfect generation by the model, as shown by our ablation studies (Section 5.3 and Appendix B).
3. For each relation proposed in the output, we search for the closest entity that exactly matches the predicted tail entity. If such an entity does not exist, the relation is discarded.
4. We discard entities or relations whose predicted type does not belong to the dataset-dependent list of types.

To better explain the DP alignment in step 2, consider the example below where the output contains a misspelled entity word, "Aciclovir" (instead of "acyclovir"). The cleaned output containing the word "Aciclovir", tokenized as "A-cicl-o-vir", is matched to "a-cycl-o-vir" in the input, from which we deduce that it refers to "acyclovir".

> **Generated output:** Six days after starting [ Aciclovir | *drug* ] she exhibited signs of [ [ lithium | *drug* ] toxicity | *disease* | *effect* = Aciclovir | *effect* = lithium ].
>
> **Cleaned output:** Six days after starting Aciclovir she exhibited signs of lithium toxicity .

**Multi-task learning.** Our method naturally allows us to train a single model on multiple datasets that can cover many structured prediction tasks. In this setting, we add the dataset name followed by the task separator **:** (for example, "ade **:**") as a prefix to each input sentence.

**Categorical prediction tasks.** For tasks such as relation prediction, where there is a limited number of valid outputs, an alternative way to perform classification is to compute class scores of all possible outputs and predict the class with the highest score. We demonstrate that we can use the output sequence likelihood as a proxy for such score. This method offers a more robust way to perform the

evaluation in low resource scenarios where generation can be imperfect (see Appendix A.3). This approach is similar to the method proposed by dos Santos et al. (2020) for ranking with language models.

## 4 STRUCTURED PREDICTION TASKS

**Joint entity and relation extraction.** Format and details for this task are provided in Section 3.

**Named entity recognition (NER).** This is an entity-only particular case of the previous task.

**Relation classification.** For this task, we are given an input sentence with *head* and *tail* entities and seek to classify the type of relation between them, choosing from a predefined set of relations. Since the head entity does not necessarily precede the tail entity in the input sentence, we add a phrase "The relationship between [ head ] and [ tail ] is" after the original input sentence. The output repeats this phrase, followed by the relation type. In the following example, the head and tail entities are "Carmen Melis" and "soprano" which have a *voice type* relation.

> **Input:** Born in Bologna, Orlandi was a student of the famous Italian [ soprano ] and voice teacher [ Carmen Melis ] in Milan. The relationship between [ Carmen Melis ] and [ soprano ] is
> **Output:** relationship between [ Carmen Melis ] and [ soprano ] = *voice type*

**Semantic role labeling (SRL).** Here we are given an input sentence along with a *predicate*, and seek to predict a list of arguments and their types. Every argument corresponds to a span of tokens that correlates with the predicate in a specific manner (e.g. subject, location, or time). The predicate is marked in the input, whereas arguments are marked in the output and are assigned an argument type. In the following example, "sold" is the predicate of interest.

> **Input:** The luxury auto maker last year [ sold ] 1,214 cars in the U.S.
> **Output:** [ The luxury auto maker | *subject* ] [ last year | *temporal* ] sold [ 1,214 cars | *object* ] [ in the U.S. | *location* ]

**Event extraction.** This task requires extracting (1) event triggers, each indicating the occurrence of a real-world event and (2) trigger arguments indicating the attributes associated with each trigger. In the following example, there are two event triggers, "attacked" of type *attack* and "injured" of type *injury*. We perform trigger detection using the same format as in NER, as shown below. To perform argument extraction, we consider a single trigger as input at a time. We mark the trigger (with its type) in the input, and we use an output format similar to joint entity and relation extraction. Below, we show an argument extraction example for the trigger "attacked", where two arguments need to be extracted, namely, "Two soldiers" of type *target* and "yesterday" of type *attack time*.

> **Trigger extraction input:** Two soldiers were attacked and injured yesterday.
> **Trigger extraction output:** Two soldiers were [ attacked | *attack* ] and [ injured | *injury* ] yesterday.
> **Argument extraction input:** Two soldiers were [ attacked | *attack* ] and injured yesterday.
> **Argument extraction output:** [ Two soldiers | *individual* | *target* = attacked ] were attacked and injured [ yesterday | *time* | *attack time* = attacked ].

**Coreference resolution.** This is the task of grouping individual text spans (*mentions*) referring to the same real-world entity. For each mention that is not the first occurrence of a group, we reference with the first mention. In the following example, "his" refers to "Barack Obama" and is marked as [ his | *Barack Obama* ] in the output.

> **Input:** Barack Obama nominated Hillary Rodham Clinton as his secretary of state on Monday. He chose her because she had foreign affairs experience as a former First Lady.
> **Output:** [ Barack Obama ] nominated [ Hillary Rodham Clinton ] as [ his | *Barack Obama* ] [ secretary of state | *Hillary Rodham Clinton* ] on Monday. [ He | *Barack Obama* ] chose [ her | *Hillary Rodham Clinton* ] because [ she | *Hillary Rodham Clinton* ] had foreign affairs experience as a former [ First Lady | *Hillary Rodham Clinton* ].

**Dialogue state tracking (DST).** Here we are given as input a history of dialogue turns, typically between a user (trying to accomplish a goal) and an agent (trying to help the user). The desired output is the dialogue state, consisting of a value for each key (or *slot name*) from a predefined list. In the input dialogue history, we add the prefixes "[ user ] :" and "[ agent ] :" to delineate user and agent turns, respectively. Our output format consists of a list of all slot names with their predicted

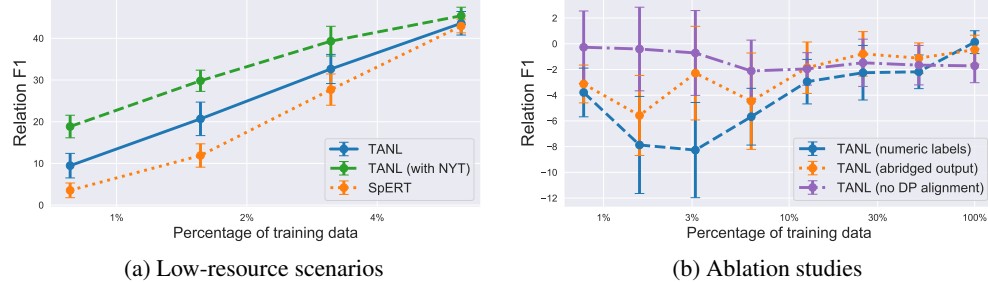

(a) Low-resource scenarios                    (b) Ablation studies

Figure 2: Experiments on the CoNLL04 dataset. (a) Our model outperforms the previous state-of-the-art model SpERT, in low-resource scenarios. (b) Ablation studies where we remove label semantics (numeric labels), augmented natural language format (abridged output) or dynamic programming alignment (no DP alignment), and plot the score difference with the non-ablated TANL.

values. We add "[ belief ]" delimiters to help the model know when to stop generating the output sequence. We tag slots that are not mentioned in the dialogue history with the value "not given" (we do not show them in the example below, for brevity).

**Input:** [ user ] **:** I am looking for a cheap place to stay [ agent ] **:** How long? [ user ] **:** Two
**Output:** [ belief ] *hotel price range* cheap, *hotel type* hotel, *duration* two [ belief ]

## 5 EXPERIMENTS

In this section, we show that our TANL framework, with the augmented natural languages outlined in Section 4, can effectively solve the structured prediction tasks considered and exceeds the previous state of the art on multiple datasets.

All our experiments start from a pre-trained T5-base model (Raffel et al., 2019). To keep our framework as simple as possible, hyperparameters are the same across all experiments, except for some dataset-specific ones, such as the maximum sequence length. Details about the experimental setup, datasets, and baselines are described in Appendix A.

### 5.1 SINGLE-TASK AND MULTI-TASK EXPERIMENTS

We use three data settings in our experiments: (1) single dataset, (2) multiple datasets for the same task (multi-dataset), and (3) all datasets across all tasks (multi-task). Table 1 shows the results.[1]

With the single-task setup, we achieve state-of-the-art performance on the following datasets: ADE, NYT, and ACE2005 (joint entity and relation extraction), FewRel and TACRED (relation classification), CoNLL-2005 and CoNLL-2012 (semantic role labeling). For example, we obtain a +6.2 absolute improvement in F1 score on the NYT dataset over the previous state of the art. Interestingly, this result is higher than the performance of models that use ground-truth entities to perform relation extraction, such as REDN (Li & Tian, 2020), which achieves a relation F1 score of 89.8. In coreference resolution, TANL performs similarly to previous approaches that employ a BERT-base model, except for CorefQA (Wu et al., 2020). To the best of our knowledge, ours is the first end-to-end approach to coreference resolution not requiring a separate mention proposal module and not enforcing a maximum mention length.

For other datasets, we obtain a competitive performance within a few points of the best baselines. We highlight that our approach uses a single model architecture that can be trained to perform *any* of the tasks without model modification. This is in stark contrast with typical discriminative models, which tend to be task-specific, as can be seen from Table 1.

In fact, under this unified framework, a single model can be trained to perform multiple or all tasks at once, with the performance being on par or even better than the single-task setting. In particular, when the dataset sizes are small such as in ADE or CoNLL04, we obtain sizable improvements and

---

[1]We are grateful to Wenxuan Zhou and Tianyu Gao for pointing out an inconsistency in computing results on the TACRED dataset, which have been corrected in Table 1.

Table 1: Results on all tasks. All numbers indicate F1 scores except noted otherwise. Datasets marked with an asterisk (*) have nested entities.

**Entity Relation Extr.**

| | CoNLL04 | | ADE* | | NYT | | ACE2005 | |
|---|---|---|---|---|---|---|---|---|
| | Entity | Rel. | Entity | Rel. | Entity | Rel. | Entity | Rel. |
| SpERT (Eberts & Ulges, 2019) | 88.9 | 71.5 | 89.3 | 78.8 | | | | |
| DyGIE (Luan et al., 2019) | | | | | | | 88.4 | 63.2 |
| MRC4ERE (Zhao et al., 2020) | 88.9 | 71.9 | | | | | 85.5 | 62.1 |
| RSAN (Yuan et al., 2020) | | | | | | 84.6 | | |
| **TANL** | 89.4 | 71.4 | 90.2 | 80.6 | **94.9** | **90.8** | **88.9** | **63.7** |
| **TANL** (multi-dataset) | 89.8 | **72.6** | 90.0 | 80.0 | 94.7 | 90.5 | 88.2 | 62.5 |
| **TANL** (multi-task) | **90.3** | 70.0 | **91.2** | **83.8** | 94.7 | 90.7 | | |

**NER**

| | CoNLL03 | OntoNotes | GENIA* | ACE2005* |
|---|---|---|---|---|
| BERT-MRC (Li et al., 2019a) | 93.0 | 91.1 | **83.8** | **86.9** |
| BERT-MRC+DSC (Li et al., 2019b) | 93.3 | **92.1** | | |
| Cloze-CNN (Baevski et al., 2019) | **93.5** | | | |
| GSL (Athiwaratkun et al., 2020) | 90.7 | 90.2 | | |
| **TANL** | 91.7 | 89.8 | 76.4 | 84.9 |
| **TANL** (multi-dataset) | 92.0 | 89.8 | 75.9 | 84.4 |
| **TANL** (multi-task) | 91.7 | 89.4 | 76.4 | |

**Relation Class.**

| | TACRED | FewRel 1.0 (validation) | | | |
|---|---|---|---|---|---|
| | | 5-way 1-shot | 5-way 5-shot | 10-way 1-shot | 10-way 5-shot |
| BERT-EM (Soares et al., 2019) | 70.1 | 88.9 | | 82.8 | |
| BERT$_{EM}$+MTB (Soares et al., 2019) | 71.5 | 90.1 | | **83.4** | |
| DG-SpanBERT (Chen et al., 2020) | 71.5 | | | | |
| BERT-PAIR (Gao et al., 2019) | | 85.7 | 89.5 | 76.8 | 81.8 |
| **TANL** | **71.9** | **94.0 ± 4.1** | **96.4 ± 4.2** | 82.6 ± 4.5 | **88.2 ± 5.9** |
| **TANL** (multi-task) | 69.1 | | | | |

**SRL**

| | CoNLL05 WSJ | CoNLL05 Brown | CoNLL2012 |
|---|---|---|---|
| Dep and Span (Li et al., 2019d) | 86.3 | 76.4 | 83.1 |
| BERT SRL (Shi & Lin, 2019) | 88.8 | 82.0 | 86.5 |
| **TANL** | 89.3 | 82.0 | **87.7** |
| **TANL** (multi-dataset) | **89.4** | **84.3** | 87.6 |
| **TANL** (multi-task) | 89.1 | 84.1 | 87.7 |

**Event Extr.**

| | ACE2005 | | | |
|---|---|---|---|---|
| | Trigger Id. | Trigger Cl. | Argument Id. | Argument Cl. |
| J3EE (Nguyen & Nguyen, 2019) | 72.5 | **69.8** | **59.9** | 52.1 |
| DyGIE++ (Wadden et al., 2019) | | 69.7 | 55.4 | **52.5** |
| **TANL** | **72.9** | 68.4 | 50.1 | 47.6 |
| **TANL** (multi-task) | 71.8 | 68.5 | 48.5 | 48.5 |

**Coreference Res.**

| | CoNLL-2012* (BERT-base ⦙ BERT-large) | | | | | | | |
|---|---|---|---|---|---|---|---|---|
| | MUC | | B$^3$ | | CEAF$_{\phi_4}$ | | Avg. F1 | |
| Higher-order c2f-coref (Lee et al., 2018) | 80.4 | | 70.8 | | 67.6 | | 73.0 | |
| SpanBERT (Joshi et al., 2020) | | 85.3 | | 78.1 | | 75.3 | | 79.6 |
| BERT+c2f-coref (Joshi et al., 2019) | 81.4 | 83.5 | 71.7 | 75.3 | 68.8 | 71.9 | 73.9 | 76.9 |
| CorefQA+SpanBERT (Wu et al., 2020) | **86.3** | **88.0** | **77.6** | **82.2** | **75.8** | **79.1** | **79.9** | **83.1** |
| **TANL** | 81.0 | | 69.0 | | 68.4 | | 72.8 | |
| **TANL** (multi-task) | 78.7 | | 65.7 | | 63.8 | | 69.4 | |

**DST**

| | MultiWOZ 2.1 (Joint Accuracy) |
|---|---|
| TRADE (Wu et al., 2019) | 45.6 |
| SimpleTOD (Hosseini-Asl et al., 2020) | **55.7** |
| **TANL** | 50.5 |
| **TANL** (multi-task) | 51.4 |

become the new state of the art (from 80.6 to 83.7 for ADE relation F1, and from 89.4 to 90.6 for CoNLL04 entity F1). The only case where our multi-task model has notably lower scores is coreference resolution, where the input documents are much longer than in the other tasks. Since the maximum sequence length in the multi-task experiment (512 tokens) is smaller than in the single-dataset coreference experiment (1,536 tokens for input and 2,048 for output), the input documents need to be split into smaller chunks, and this hurts the model's ability to connect multiple mentions of the same entity across different chunks. From the multi-task experiment, we leave out all datasets based on ACE2005 except for event extraction due to overlap between train and test splits for different tasks. We discuss our experiments in more detail in Appendix A.

All results presented in this paper are obtained from a pre-trained T5-base model. In principle, any pre-trained generative language model can be used, such as BART (Lewis et al., 2020) or GPT-2 (Radford et al., 2019). It would be interesting to check whether these models are as capable as T5 (or even better) at learning to translate between our augmented languages. We leave this as a direction for future investigation.

## 5.2 LOW-RESOURCE SETTINGS

Multiple experiments suggest that TANL is data-efficient compared to other baselines. On the FewRel dataset, a benchmark for few-shot relation classification, our model outperforms the best baselines BERT$_{EM}$ and BERT$_{EM}$+MTB (Devlin et al., 2019; Soares et al., 2019), where the MTB version uses a large entity-linked text corpus for pre-training. On the TACRED relation classification dataset, our model also improves upon the best baselines (from 71.5 to 71.9). While TACRED is not specifically a few-shot dataset, we observe that there are many label types that rarely appear in the training set, some of them having less than 40 appearances out of approximately 70,000 training label instances. We show the occurrence statistics for all label types in the appendix (Table 3), demonstrating that the dataset is highly imbalanced. Nonetheless, we find that our model performs well, even on instances involving scarce label types. This ability distinguishes our models from other few-shot approaches such as prototypical networks (Snell et al., 2017) or matching networks (Vinyals et al., 2016), which are designed only for few-shot scenarios but do not scale well on real-world data which often contains a mix of high and low-resource label types.

Our low-resource study on the joint entity and relation extraction task also confirms that our approach is more data-efficient compared to other methods. We experiment on the CoNLL04 dataset, using only 0.8% (9 sentences) to 6% (72 sentences) of the training data. Our approach outperforms SpERT (a state-of-the-art discriminative model for joint entity and relation extraction) in this low-resource regime, whereas the performance is similar when using the full training set.

Thanks to the unified framework, we can easily train on a task, potentially with larger resources, and adapt to other low-resource end tasks (transfer learning). To show this, we train a model with a large dataset from joint entity and relation extraction (NYT) and fine-tune it on a limited portion of the CoNLL04 dataset (Figure 2), obtaining a significant increase in performance (up to +9 relation F1).

Finally, in Appendix C we analyze how the size of the training dataset affects the number of generation errors of our model.

## 5.3 ABLATION STUDIES

We conduct ablation studies to demonstrate that label semantics, augmented natural language format, and optimal alignment all contribute to the effectiveness of TANL (Figure 2b). Further details on these ablation studies can be found in Appendix B.

**Numeric labels:** To prevent the model from understanding the task through label semantics, we use numeric labels. This substantially hurts the performance, especially in a low-resource setting where transfer learning is more important. **Abridged output:** Second, to determine the impact of the augmented natural language format outlined in Section 4, we experiment with a format which does not repeat the entire input sentence. We find that this abridged format consistently hurts model performance, especially in low-resource scenarios. In other tasks, we generally find that a more natural-looking format usually performs better (see Appendix A.3). **No DP alignment:** We use exact word matching instead of the dynamic programming alignment described in Section 3.

## 6   DISCUSSION AND CONCLUSION

We have demonstrated that our unified text-to-text approach to structured prediction can handle all the considered tasks within a simple framework and offers additional benefits in low-resource settings. Unlike discriminative models common in the literature, TANL is generative as it translates from an input to an output in augmented natural languages. These augmented languages are flexible and can be designed to handle a variety of tasks, some of which are complex and previously required sophisticated prediction modules. By streamlining all tasks to be compatible with a single model, multi-task learning becomes seamless and yields state-of-the-art performance for many tasks.

Generative models, and in particular sequence-to-sequence models, have been used successfully in many NLP problems such as machine translation, text summarization, etc. These tasks involve mappings from one *natural* language input to another *natural* language output. However, the use of sequence modeling for structured prediction has received little consideration. This is perhaps due to the perception that the generative approach is too unconstrained and that it would not be a robust way to generate a precise output format that corresponds to structured objects, or that it may add an unnecessary layer of complexity with respect to discriminative models. We demonstrate that this is quite the opposite. The generative approach can easily handle disparate tasks, even at the same time, by outputting specific structures appropriate for each task with little, if any, format error.

We note that one drawback of the current generative approach is that the time complexity for each token generation is $\mathcal{O}(L^2)$ where $L$ is the sentence length. However, there have been recent advances in the attention mechanism that reduce the complexity to $\mathcal{O}(L \log L)$ as in Reformer (Kitaev et al., 2020), or to $\mathcal{O}(L)$ as in Linformer (Wang et al., 2020). Incorporating these techniques in the future can significantly reduce computation time and allow us to tackle more complex tasks, as well as improve on datasets with long input sequences such as in coreference resolution.

Based on our findings, we believe that generative modeling is highly promising but has been an understudied topic in structured prediction. Our findings corroborate a recent trend where tasks typically treated with discriminative methods have been successfully solved using generative approaches (Brown et al., 2020; Izacard & Grave, 2020; Schick & Schütze, 2020). We hope our results will foster further research in the generative direction.

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

## A  EXPERIMENTAL SETUP, DATASETS, AND BASELINES

In all experiments, we fine-tune a pre-trained T5-base model (Raffel et al., 2019), to exploit prior knowledge of the natural language. The family of T5 models was specially designed for downstream text-to-text tasks, making them suitable for our needs. The T5-base model has about 220 million parameters. For comparison, both encoder and decoder are similar in size to BERT-base (Devlin et al., 2019). We use the implementation of HuggingFace's Transformers library (Wolf et al., 2019).

To keep our framework as simple as possible, hyperparameters are the same across the majority of our experiments. We use: 8 V100 GPUs with a batch size of 8 per GPU; the AdamW optimizer (Kingma & Ba, 2015; Loshchilov & Hutter, 2019); linear learning rate decay starting from 0.0005; maximum input/output sequence length equal to 256 tokens at training time (longer sequences are truncated), except for relation classification, coreference resolution, and dialogue state tracking (see below). The number of fine-tuning epochs is adjusted depending on the size of the dataset, as described later. With these settings, one fine-tuning step takes approximately 0.8 seconds. This translates into 15 seconds per epoch for the (relatively small) CoNLL04 dataset (joint entity-relation extraction) and 16 minutes per epoch for the (much larger) OntoNotes dataset (NER). At inference time, we employ beam search with 8 beams, and we adjust the maximum sequence length depending on the length of the sentences in each dataset. Note that beam search is not an essential part of our framework, as we find that greedy decoding gives almost identical results.

In the rest of this section, we describe datasets and baselines for each structured prediction task, as well as additional insights on particular experiments. Results of all experiments are given in Table 1. Unless otherwise specified, micro-F1 scores are reported. Most experiments are run more than once, as described below, and the average result is reported. Table 5 shows input-output examples from different datasets.

For the multi-task experiment, we train for 50 epochs on 80 GPUs, with a batch size of 3 per GPU. The maximum input/output sequence length is set to 512 for all tasks.

### A.1  JOINT ENTITY-RELATION EXTRACTION

**Datasets.**  We experiment on the following datasets: CoNLL04 (Roth & Yih, 2004), ADE (Gurulingappa et al., 2012), NYT (Riedel et al., 2010), and ACE2005 (Walker et al., 2006).

- The **CoNLL04** dataset consists of sentences extracted from news articles, with four entity types (*location*, *organization*, *person*, *other*) and five relation types (*work for*, *kill*, *organization based in*, *live in*, *located in*). As in previous work, we use the training (922 sentences), validation (231 sentences), and test (288 sentences) split by Gupta et al. (2016). We train for 200 epochs and report our test results averaged over 10 runs.

- The **ADE** dataset consists of $4,272$ sentences extracted from medical reports, with two entity types (*drug*, *disease*) and a single relation type (*effect*). This dataset has sentences with nested entities. As in previous work, we conduct a 10-fold cross-validation and report the average macro-F1 results across all 10 splits (except for the multi-task experiment, which is carried out once and uses the first split of the ADE dataset). We train for 200 epochs.

- The **NYT** dataset (Zeng et al., 2018) is based on the New York Times corpus and was automatically labeled with distant supervision by Riedel et al. (2010). We use the preprocessed version of Yu et al. (2019). This dataset has three entity types (*location*, *organization*, *person*) and 24 relation types (such as *place of birth*, *nationality*, *company*). It consists of 56,195 sentences for training, 5,000 for validation, and 5,000 for testing. We train for 50 epochs and report our test results averaged over 5 runs.

- The **ACE2005** dataset is derived from the ACE2005 corpus (Walker et al., 2006) and consists of sentences from a variety of domains, including news and online forums. We use the processing code of Luan et al. (2019). After filtering out the sentences without entities, we get 7,477 sentences for training, 1789 for validation, and 1517 for testing. It has seven entity types (*location*, *organization*, *person*, *vehicle*, *geographical entity*, *weapon*, *facility*) and six relation types (*PHYS*, *ART*, *ORG-AFF*, *GEN-AFF*, *PER-SOC*, *PART-WHOLE*).

Table 2: Details about the single-dataset experiments in joint entity-relation extraction and named entity recognition.

| Dataset | # Epochs | # Runs | Results | |
|---|---|---|---|---|
| *Joint entity-relation extraction* | | | Entity F1 | Relation F1 |
| CoNLL04 | 200 | 10 | $89.4 \pm 0.3$ | $71.4 \pm 1.1$ |
| ADE | 200 | 10 | $90.2 \pm 0.7$ | $80.6 \pm 1.5$ |
| NYT | 50 | 5 | $94.9 \pm 0.1$ | $90.8 \pm 0.1$ |
| ACE2005 | 100 | 10 | $88.9 \pm 0.1$ | $63.7 \pm 0.7$ |
| *Named entity recognition* | | | Entity F1 | |
| CoNLL03 | 50 | 10 | $91.7 \pm 0.1$ | |
| OntoNotes | 20 | 10 | $89.8 \pm 0.1$ | |
| GENIA | 50 | 10 | $76.4 \pm 0.4$ | |
| ACE2005 | 50 | 10 | $84.9 \pm 0.2$ | |

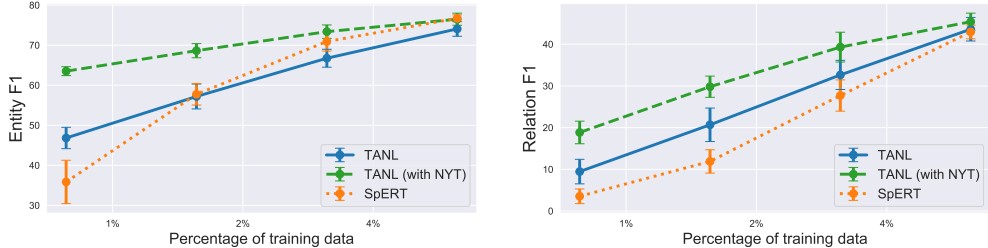

Figure 3: Low-resource experiments on the CoNLL04 dataset.

The natural labels we use for the relation types are: *physical*, *artifact*, *employer*, *affiliation*, *social*, *part of*. We train for 100 epochs and report our test results averaged over 10 runs.

For all single-dataset experiments, Table 2 shows the number of training epochs, the number of runs, and the standard deviations, in addition to the average results, which are already reported in Table 1.

**Baselines.** SpERT (Eberts & Ulges, 2019) is a BERT-based model which performs span classification and then relation classification. Multi-turn QA (Li et al., 2019c) casts the problem as a multi-turn question answering task. ETL-Span (Yu et al., 2019) uses BiLSTM and decomposes the problem into two tagging sub-problems: head entity extraction, and tail entity and relation extraction. WDec (Nayak & Ng, 2020) uses an encoder-decoder architecture to directly generate a list of relation tuples. MRC4ERE (Zhao et al., 2020) improves on the question answering approach by leveraging a diverse set of questions. RSAN (Yuan et al., 2020) is a sequence labeling approach which utilizes a relation-aware attention mechanism.

**Low-resource experiments.** As outlined in Section 5.2, we experiment on the CoNLL04 dataset with only a limited portion of the training set available and plot our results in Figure 3. Comparison is made with SpERT (Eberts & Ulges, 2019), a state-of-the-art discriminative model. TANL performs better than SpERT with fewer data, especially on the more complex task of relation extraction (right plot). We also show our method's performance with preliminary fine-tuning on the NYT dataset for one epoch, which significantly improves the performance on both entity and relation extraction. To account for the small dataset size, we fine-tune on CoNLL04 for 2,000 epochs ($10\times$ the number of epochs we use to train on the full CoNLL04 dataset). For a fair comparison, we train SpERT for 20, 200, and 2000 epochs (respectively $1\times$, $10\times$, and $100\times$ the number of epochs suggested in the paper), and report the best result among the three, which is always obtained with 200 epochs. We plot mean and standard deviation over 10 runs (each model being fine-tuned on the same 10 subsets of the training set and evaluated on the entire test set). For reference, the smallest training set has only 9 sentences (0.8% of the total), effectively consisting in a few-shot learning scenario.

**Multi-dataset experiments.** We train a single model on all four datasets for 20 epochs and report the average over 10 runs. We use a different split of the ADE dataset in each run.

## A.2 Named entity recognition

**Datasets.** We experiment on two flat NER datasets, CoNLL03 (Sang & Meulder, 2003) and OntoNotes (Pradhan et al., 2013), and two nested NER datasets, GENIA (Ohta et al., 2002) and ACE2005 (Walker et al., 2006).

- For the **CoNLL03** dataset (Sang & Meulder, 2003) we use the same processing and splits as Li et al. (2019a), resulting in 14,041 sentences for training, 3,250 for validation, and 3,453 for testing. This dataset has four entity types (*location*, *organization*, *person*, *miscellaneous*). We train for 50 epochs and report our test results averaged over 10 runs.

- The English **OntoNotes** dataset (Pradhan et al., 2013) consists of 59,924 sentences for training, 8,528 for validation, and 8,262 for testing. It has 18 entity types (such as *person*, *organization*, *date*, *percent*). We train for 20 epochs and report our test results averaged over 10 runs.

- The **GENIA** dataset (Ohta et al., 2002) consists of sentences from the molecular biology domain. As in previous work, we use the processing and splits of Finkel & Manning (2009) resulting in 14,824 sentences for training, 1,855 for validation, and 1,854 for testing. There are five entity types (*protein*, *DNA*, *RNA*, *cell line*, *cell type*). We train for 50 epochs and report our test results averaged over 10 runs.

- The **ACE2005** dataset for nested NER is based on the ACE2005 corpus (Walker et al., 2006), but is different from the one used for joint entity-relation extraction. We use the same processing and splits of Li et al. (2019a), resulting in 7,299 sentences for training, 971 for validation, and 1,060 for testing. It has the same seven entity types as the ACE2005 dataset used for joint entity-relation extraction. We train for 50 epochs and report our test results averaged over 10 runs.

As for joint entity-relation extraction, Table 2 summarizes our setup and results (with standard deviations) for the single-dataset experiments.

**Baselines.** State-of-the-art results on popular NER datasets are mostly detained by BERT-MRC (Li et al., 2019a) and BERT-MRC + DSC (Li et al., 2019b), which formulate the problem as a machine reading comprehension task, solved by asking multiple questions. ClozeCNN (Baevski et al., 2019) leverages a cloze-driven pre-training. Seq2seq-BERT (Straková et al., 2019) uses a seq2seq model to output the list of entity types. Second-best learning and decoding (Shibuya & Hovy, 2019) iteratively decodes nested entities starting from the outermost ones, using the Viterbi algorithm. For flat NER, our approach is similar to GSL (Athiwaratkun et al., 2020).

**Multi-dataset experiments.** We train a single model on all four datasets for 10 epochs and report our results averaged over 5 runs.

## A.3 Relation classification

**Datasets.** We experiment on FewRel (Han et al., 2018) and TACRED (Zhang et al., 2017).

- **FewRel** consists of 100 relations with 7 instances for each relation. The standard evaluation for this benchmark uses few-shot $N$-way $K$-shot settings, which we follow. The entire dataset is split into train (64 relations), validation (16 relations) and test set (20 relations). We train our model on the meta-training set, which has no overlapping classes with the evaluation set. At evaluation time, given a support set and a query set on a new task, we fine-tune the model on the support set to learn the new task and evaluate on the query set.

- **TACRED** is a large-scale relation classification dataset with 106,344 examples (68,164 for training, 22,671 for validation, and 15,509 for testing), covering 41 relation types. We train for 5 epochs and report our test results averaged over 5 runs. The maximum input sequence length is set to 300, whereas the maximum output sequence length is set to 64 during training and 128 during inference.

**Baselines.** We compare our approach with the following two models in the literature. The first is BERT-pair (Gao et al., 2019), a sequence classification model based on BERT, which learns to optimize the scores indicating the relation between a query instance and other supporting instances

for the same relation. The second is BERT$_{EM}$ + Matching the Blanks (MTB) (Soares et al., 2019). BERT$_{EM}$ uses entity markers indicating the start and the end of the head and tail entities in the input sentence. MTB is a pre-training based on an additional large corpus of relation data. Nevertheless, our model is able to outperform BERT$_{EM}$+MTB in certain cases, such as the 5-way 1-shot setting on FewRel.

**Augmented natural language formats.** We experiment with many augmented natural language formats, as shown below:

**Input (chosen):** Born in Bologna, Orlandi was a student of the famous Italian [ soprano ] and voice teacher [ Carmen Melis ] in Milan. The relationship between [ Carmen Melis ] and [ soprano ] is

**Output (chosen) :** relationship between [ Carmen Melis ] and [ soprano ] = *voice type*

**Input (alternative 1):** Born in Bologna, Orlandi was a student of the famous Italian [ soprano ] and voice teacher [ Carmen Melis ] in Milan. The relationship between [ Carmen Melis ] and [ soprano ] is

**Output (alternative 1):** *voice type*

**Input (alternative 2):** Born in Bologna, Orlandi was a student of the famous Italian [ soprano | *tail* ] and voice teacher [ Carmen Melis | *head* ] in Milan.

**Output (alternative 2):** relationship between [ Carmen Melis ] and [ soprano ] = *voice type*

The alternative 1 version has a shorted output which only produces the keyword such as *voice type* corresponding to the predicted relation. However, we find that it does not perform as well as the chosen format. We hypothesize that it is due to the rich semantics of the sentence "relationship between [ Carmen Melis ] and [ soprano ]", and possibly softer gradient information on the longer sequence which improves training.

The alternative 2 version annotates the head vs. tail information for the entities directly in the input, instead of using a phrase such as "relationship between [ Carmen Melis ] and [ soprano ]" to specify that "Carmen Melis" is the head entity. However, this format also does not perform as well, possibly because the meaning of the words *head* and *tail* are not fully understood in this context. Overall, the chosen format sounds the most natural out of all options and is closer to natural language, which we use as our guiding principle to design our augmented natural language formats.

**TACRED results and label sparsity.** A major factor for our state-of-the-art result on the TACRED dataset is the shared semantics across different labels, which is particularly beneficial in the case of sparse labels. In Table 3 we show the relation types in natural words, the number of training examples, which can be quite small, and the test recall (*i.e.*, out of all ground truth relations for a given type, how many we predict correctly). We can see that even though some relation types such as *date of birth* have as little as 64 labels in the training set (less than 0.1% of the entire set), our model is able to correctly predict this relation type with recall 77.8%.

The ability to handle few-shot cases allows our approach to perform well in real-world data such as TACRED, where the labels can be highly imbalanced. As seen in Table 3, only a few instances such as *employee of*, *top members employees*, *title*, and *no relation* dominate the majority of the training set (approximately 60,000 out of 68,000), where the rest can be considered scarce. Our model is different from other approaches specifically designed for few-shot scenarios in that it scales across different levels of data.

**Few-shot experiments.** For the FewRel dataset, we perform meta-training by training the model on the training set of FewRel for 1 epoch. During evaluation, we fine-tune the model on the support set for each episode for 2,500 epochs in the 1-shot cases, and for 500 epochs in the 5-shot cases.

**Likelihood-based prediction.** In relation classification, we aim to predict one class out of a pre-defined set of classes, so we can perform prediction by using sequence likelihoods as class scores. This helps improve the performance particularly in the case of few-shot scenarios, where the generation of label types can be imperfect since the model has seen only one or few instances of each type. With the likelihood prediction, we obtain a slight improvement across the board. For instance, we improve from an F1 score of 95.6 ± 4.8 to 96.4 ± 4.2 for the 5-way 5-shot case of FewRel. For TACRED, using the likelihood approach yields a smaller improvement, possibly due to the fact that the model can generate exact label types given enough training resources, unlike in the few-shot case. All our reported numbers on the FewRel and TACRED datasets are obtained by using this approach.

Table 3: TACRED recall by relation type (in one of our 5 runs), with number of training, validation, and test examples.

| Relation type | # Train | # Dev | # Test | Test recall |
|---|---|---|---|---|
| country of death | 7 | 47 | 9 | 44.4 |
| dissolved | 24 | 9 | 2 | 50.0 |
| country of birth | 29 | 21 | 5 | 0.0 |
| state or province of birth | 39 | 27 | 8 | 50.0 |
| state or province of death | 50 | 42 | 14 | 64.3 |
| religion | 54 | 54 | 47 | 48.9 |
| date of birth | 64 | 32 | 9 | 77.8 |
| city of birth | 66 | 34 | 5 | 20.0 |
| charges | 73 | 106 | 103 | 85.4 |
| number of employees members | 76 | 28 | 19 | 63.2 |
| shareholders | 77 | 56 | 13 | 0.0 |
| city of death | 82 | 119 | 28 | 32.1 |
| founded | 92 | 39 | 37 | 83.8 |
| political religious affiliation | 106 | 11 | 10 | 40.0 |
| website | 112 | 87 | 26 | 88.5 |
| cause of death | 118 | 169 | 52 | 36.5 |
| member of | 123 | 32 | 18 | 0.0 |
| founded by | 125 | 77 | 68 | 82.4 |
| date of death | 135 | 207 | 54 | 55.6 |
| schools attended | 150 | 51 | 30 | 73.3 |
| siblings | 166 | 31 | 55 | 76.4 |
| members | 171 | 86 | 31 | 48.4 |
| other family | 180 | 81 | 60 | 46.7 |
| children | 212 | 100 | 37 | 67.6 |
| state or province of headquarters | 230 | 71 | 51 | 80.4 |
| spouse | 259 | 160 | 66 | 69.7 |
| subsidiaries | 297 | 114 | 44 | 45.5 |
| origin | 326 | 211 | 132 | 62.1 |
| state or provinces of residence | 332 | 73 | 81 | 54.3 |
| cities of residence | 375 | 180 | 189 | 58.7 |
| city of headquarters | 383 | 110 | 82 | 70.7 |
| age | 391 | 244 | 200 | 96.0 |
| parents | 439 | 153 | 150 | 64.7 |
| countries of residence | 446 | 227 | 148 | 37.8 |
| country of headquarters | 469 | 178 | 108 | 60.2 |
| alternate names | 913 | 377 | 224 | 89.3 |
| employee of | 1,525 | 376 | 264 | 70.1 |
| top members employees | 1,891 | 535 | 346 | 84.4 |
| title | 2,444 | 920 | 500 | 86.8 |
| no relation | 55,113 | 17,196 | 12,184 | 93.8 |

## A.4 SEMANTIC ROLE LABELING

**Datasets.** We use CoNLL-2005 (Carreras & Màrquez, 2005) and the CoNLL-2012 English subset of OntoNotes 5.0 (Pradhan et al., 2013) in our experiments. See also Carreras & Màrquez (2005); Pradhan et al. (2012). These tasks have highly specific label types, and their natural words might be cumbersome for training. Therefore, we use the raw label types from the original datasets as presented below.

- **CoNLL-2005** focuses on the semantic roles given verb predicates. The argument notation is the following. V: verb; A0: acceptor; A1: thing accepted; A2: accepted from; A3: attribute; AM-MOD: modal; AM-NEG: negation.

- **CoNLL-2012**. The argument notation, taken from Pradhan et al. (2012), is as follows.

  Numbered arguments (A0-A5, AA): Arguments defining verb-specific roles. Their semantics depends on the verb and the verb usage in a sentence, or verb sense. The most frequent roles are A0 and A1. Commonly, A0 stands for the agent, and A1 corresponds to the patient or theme of the proposition. However, no consistent generalization can be made across different verbs or different senses of the same verb. PropBank takes the definition of verb senses from VerbNet, and for each verb and each sense defines the set of possible roles for that verb usage, called the roleset. The definition of rolesets is provided in the PropBank Frames files, made available for the shared task as an official resource to develop systems.

  Adjuncts (AM-): General arguments that any verb may take optionally. The following are the 13 types of adjuncts. AM-ADV: general-purpose; AM-CAU: cause; AM-DIR: direction; AM-DIS: discourse marker; AM-EXT: extent; AM-LOC: location; AM-MNR: manner; AM-MOD: modal verb; AM-NEG: negation marker; AM-PNC: purpose; AM-PRD: predication; AM-REC: reciprocal; AM-TMP: temporal.

  References (R-): Arguments representing arguments realized in other parts of the sentence. The role of a reference is the same as the role of the referenced argument. The label is an R-tag prefixed to the label of the referent, e.g., R-A1.

**Baselines.** We compare our results with Dependency and Span SRL (Li et al., 2019d), which uses a Bi-LSTM with highway connection and biaffine scorers, and BERT-SRL (Shi & Lin, 2019), BERT-based model which predicts the spans based on the contextual and positional embeddings.

**Multi-dataset experiments.** We train a single model on all datasets for 50 epochs and report our results averaged over 5 runs.

## A.5 EVENT EXTRACTION

**Datasets.** We use the **ACE2005** English event data (Walker et al., 2006) in our experiments, following standard event extraction literature. We use the same split as previous work (Ji & Grishman, 2008; Li et al., 2013) with 529 documents for training, 30 for validation, and 40 for testing. Since the majority of event triggers and their corresponding arguments are within the same sentence, we perform the event extraction task only at the sentence level. We fine-tune our model for 50 epochs on this dataset.

**Baselines.** We compare our method with the following two baseline models in the literature. The first is J3EE (Nguyen & Nguyen, 2019), a Bi-GRU based model that jointly performs event trigger detection, event mention detection, and event argument classification. J3EE performs event trigger detection and event mention detection as sequence tagging problems, and event argument classification as a classification problem, given any trigger and candidate argument pair. The second baseline is DyGIE++ (Wadden et al., 2019), a BERT based multi-task learning framework for the tasks of coreference resolution, relation extraction, named entity recognition, and event extraction. DyGIE++ enumerates all possible phrases within a sentence and predicts the best entity type and trigger type for each of these phrases. Argument roles are then predicted for each trigger and entity pair.

### A.6 COREFERENCE RESOLUTION

**Datasets.** We use the standard OntoNotes benchmark defined in the **CoNLL-2012** shared task (Pradhan et al., 2012). It consists of 2,802 documents for training, 343 for validation, and 348 for testing, for a total of about one million words. Since documents can be large (up to 4,000 words), we split each document into partially overlapping chunks up to 1,024 words long (and 128 or 196 words for the multi-task experiment, respectively for training and for testing). At test time, we merge groups from different chunks if they have at least one mention in common in order to obtain document-level predictions. As in prior work, evaluation is done by computing the average F1 score of the three standard metrics for coreference resolution: MUC, $B^3$, $CEAF_{\phi_4}$. We train for 100 epochs, with a maximum sequence length equal to 1,536 tokens for input and 2,048 for output, and a batch size of 1 per GPU.

**Baselines.** The e2e-coref model (Lee et al., 2017) is among the first end-to-end approaches to coreference resolution. It considers all spans as potential mentions and learns a distribution over possible antecedents for each span. Higher-order c2f-coref (Lee et al., 2018) iteratively refines span representations taking into account higher-order relations between mentions. BERT + c2f-coref (Joshi et al., 2019) combines the previous approach with BERT. SpanBERT (Joshi et al., 2020) introduces a new pretraining method which is designed to better represent and predict spans of text. CorefQA (Wu et al., 2020) generate queries for each mention from a mention proposal network and uses a question answering framework to extract text spans of coreferences.

### A.7 DIALOGUE STATE TRACKING

**Datasets.** We use the **MultiWOZ 2.1** (Eric et al., 2020) task oriented dialogue dataset in our experiments. It consists of 8,420 conversations for training, 1,000 for validation, and 999 for testing. We follow the pre-processing procedure put forward in (Wu et al., 2019) for dialogue state tracking. In addition, we remove the "police" and "hospital" domains from the training set since they are not present in the test set. Removing these two domains reduces the training set size from 8,420 to 7,904. We fine-tune for 100 epochs, with maximum sequence length set to 512 tokens. We train a single generative model that predicts the dialogue state for the entire dialogue history up to the current turn. Following prior work, we report the joint accuracy.

**Baselines.** We compare our performance on MultiWOZ 2.1 against SimpleTOD (Hosseini-Asl et al., 2020), the current state of the art for MultiWOZ dialogue state tracking. SimpleTOD uses a sequence to sequence approach based on the GPT-2 (Radford et al., 2019) language model. Unlike our approach, SimpleTOD is trained to jointly generate actions and responses as well as dialogue states.

## B ABLATION STUDIES

As outlined in Section 5.3, we conduct ablation studies on the CoNLL04 dataset (joint entity and relation extraction) to demonstrate the importance of label semantics, natural output format, and optimal alignment. We compare TANL with the following three variations.

- Numeric labels: we use numbers (1, 2, 3, ...) to indicate entity and relation types in the output sentences, as in the following example.

  **Output:** [ Boston University | 2 ]'s [ Michael D. Papagiannis | 3 | 1 = Boston University ] said he believes the crater was created [ 100 million years | 4 ] ago when a 50-mile-wide meteorite slammed into the [ Earth | 1 ].

- Abridged output: here, the output consists of a list of entities, enclosed between [ ] tokens, without text between them.

  **Output:** [ Boston University | organization ] [ Michael D. Papagiannis | person | works for = Boston University ] [ 100 million years | other ] [ Earth | location ]

- No alignment: we process output sentences without the alignment module. For each predicted entity or relation, we look for the first exact match in the input sentence (the entity or relation is discarded if no exact match is found).

Table 4: Ablation studies on the CoNLL04 dataset (using the full training set, and using only 50% of the training sentences). We report mean and standard deviation over 10 runs.

| Model | CoNLL04 | | CoNLL04 (50%) | |
|---|---|---|---|---|
| | Entity F1 | Relation F1 | Entity F1 | Relation F1 |
| TANL | **89.44** $\pm$ 0.30 | 71.44 $\pm$ 1.15 | **87.15** $\pm$ 1.08 | **68.30** $\pm$ 1.47 |
| TANL (numeric labels) | 89.13 $\pm$ 0.45 | **71.57** $\pm$ 0.89 | 86.59 $\pm$ 0.94 | 66.12 $\pm$ 1.31 |
| TANL (abridged output) | 88.42 $\pm$ 0.67 | 70.98 $\pm$ 1.12 | 86.11 $\pm$ 0.55 | 67.18 $\pm$ 1.18 |
| TANL (no alignment) | 87.88 $\pm$ 0.31 | 69.72 $\pm$ 1.31 | 85.56 $\pm$ 1.01 | 66.64 $\pm$ 1.54 |

Figure 4: Ablation studies on CoNLL04, using different portions of the training dataset.

The outcomes of these experiments are shown in Figures 2 and 4, and Table 4. We run all experiments using a variable amount of training data, from 100% (1,153 sentences) down to 0.8% (9 sentences), and always evaluate on the entire test set (288 sentences). To account for the variable size of the training dataset, we adjust the number of training epochs as follows: 200 epochs when using all training data; 400 epochs for 50% of the training data; 800 epochs for 25%; 1,600 epochs for 12.5%; 2,000 epochs for all remaining cases (6.3%, 3.1%, 1.6%, 0.8%).

Results show that all three components (label semantics, natural output format, and alignment) positively contribute to the effectiveness of TANL. The impact of label semantics is not noticeable when using the full CoNLL04 training dataset (natural and numeric labels give similar F1 scores), but it becomes statistically relevant when using 50% of the training data, or less. On the other hand, the impact of alignment is higher when the training dataset is larger. Interestingly, for entity extraction (left plot of Figure 4), repeating the input sentence is more important than using natural labels, whereas the opposite is true for relation extraction (right plot).

From these experiments, we deduce that: (1) the model indeed uses latent knowledge about label semantics, especially when the amount of training data is low; (2) using a "natural" output format (which replicates the input sentence as much as possible) allows the model to make more accurate predictions, likely by encouraging the use of the entire input as context; (3) alignment helps in locating the correct entity spans in the input sentence, and in correcting mistakes made by the model when replicating the input.

## C  ANALYSIS OF GENERATION ERRORS

The performance of TANL crucially depends on the quality of the generated output sentences. Figure 5 shows how often the following kinds of generation errors occur on the CoNLL04 dataset.

- Reconstruction errors: the output sentence does not exactly replicate the input sentence.
- Format errors: the augmented natural language format is invalid.
- Entity errors: there is at least one relation whose predicted tail entity does not match any predicted entity.
- Label errors: there is at least one predicted entity or relation type that does not exactly match any of the dataset's possible types.

Reconstruction errors are by far the most common, but they are mitigated by our alignment step. When using the full CoNLL04 training dataset, other errors appear very infrequently; therefore, it is not necessary to add further post-processing steps to mitigate them. We perform this generation

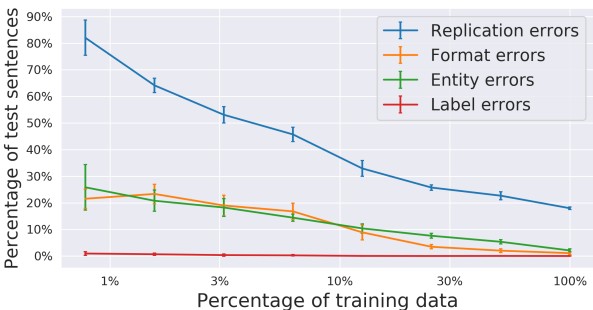

Figure 5: Percentage of output sentences presenting different kinds of errors, when training with a variable portion of the CoNLL04 training dataset.

error analysis on the CoNLL04 because it is the smallest of the benchmarks we consider, and as a result, the generation errors on CoNLL04 are likely to be the most significant. Yet when training on only a limited portion of the training data, format, and entity errors do occur. In this low-resource setting, TANL would benefit from additional post-processing. We leave the investigation of such post-processing strategies aimed at low-resource scenarios for future work.

Table 5: Input-output examples for all structured prediction datasets.

| Dataset | Input | Output |
|---|---|---|
| CoNLL04 | Boston University's Michael D. Papagiannis said he believes the crater was created 100 million years ago when a 50-mile-wide meteorite slammed into the Earth. | [ Boston University | *organization* ]'s [ Michael D. Papagiannis | *person* | works for = Boston University ] said he believes the crater was created [ 100 million years | *other* ] ago when a 50-mile-wide meteorite slammed into the [ Earth | *location* ]. |
| ADE | Progressive hypoxemia mandated endotracheal intubation 1 week after rituximab administration and led to death 4 weeks after admission. | [ Progressive hypoxemia | *disease* | *effect* = rituximab ] mandated endotracheal intubation 1 week after [ rituximab | *drug* ] administration and led to death 4 weeks after admission. |
| NYT | At the Triboro Coach depot in East Elmhurst, Queens, this morning, about 20 workers wore or carried red union bandannas and held placards with messages like, "The Mayor Lied, There Goes Your Ride" and "On Strike." | At the Triboro Coach depot in [ East Elmhurst | *location* | *neighborhood of* = Queens ], [ Queens | *location* | *contains* = East Elmhurst ], this morning, about 20 workers wore or carried red union bandannas and held placards with messages like, "The Mayor Lied, There Goes Your Ride" and "On Strike." |
| ACE2005 (entity-rel extraction) | that is the very joyous town of palestine, west virginia, on the news that jessica lynch is eventually going to come home. | [ that | *geographical entity* ] is the very joyous [ town | *geographical entity* ] of [ palestine | *geographical entity* | *part of* = west virginia ], [ west virginia | *geographical entity* ], on the news that [ jessica lynch | *person* | *located in* = home ] is eventually going to come [ home | *geographical entity* ]. |
| CoNLL03 | Charlton, 61, and his wife, Peggy, became citizens of Ireland when they formally received Irish passports from deputy Prime Minister Dick Spring who said the honour had been made in recognition of Charlton's achievements as the national soccer manager. | [ Charlton | *person* ], 61, and his wife, [ Peggy | *person* ], became citizens of [ Ireland | *location* ] when they formally received [ Irish | *miscellaneous* ] passports from deputy Prime Minister [ Dick Spring | *person* ] who said the honour had been made in recognition of [ Charlton | *person* ]'s achievements as the national soccer manager. |
| OntoNotes | The eventual court decision could become a landmark in Dutch corporate law because the lawsuit ASKO plans to file would be the first to challenge the entire principle and practice of companies issuing voting preferred shares to management - controlled trusts to dilute voting power of common stockholders. | The eventual court decision could become a landmark in [ Dutch | *nationality religious political group* ] corporate law because the lawsuit [ ASKO | *organization* ] plans to file would be the [ first | *ordinal* ] to challenge the entire principle and practice of companies issuing voting preferred shares to management - controlled trusts to dilute voting power of common stockholders. |
| GENIA | Activation of CD4 positive T cells is a primary requirement for human immunodeficiency virus (HIV) entry, efficient HIV replication, and progression to AIDS, Utilizing CD4 positive T cell lines and purified T cells from normal individuals, we have demonstrated that native envelope glycoproteins of HIV, gp 160, can induce activation of transcription factor, activated protein - 1 (AP - 1). | Activation of [ CD4 positive [ T cells | *cell type* ] | *cell type* ] is a primary requirement for human immunodeficiency virus (HIV) entry, efficient HIV replication, and progression to AIDS, Utilizing [ CD4 positive T cell lines | *cell line* ] and [ purified [ T cells | *cell type* ] | *cell type* ] from normal individuals, we have demonstrated that [ native envelope glycoproteins | protein ] of HIV, [ gp 160 | *protein* ], can induce activation of [ transcription factor, activated protein - 1 | *protein* ] ([ AP-1 | *protein* ]). |
| ACE2005 (NER) | While Starbucks does partner (airlines, airports, Barnes & Noble), most of its stores are company owned. | While [ Starbucks | *organization* ] does partner ([ airlines | *organization* ], [ airports | *facility* ], [ Barnes & Noble | *organization* ]), [ most of [ [ its | *organization* ] stores | *facility* ] | *facility* ] are [ company | *organization* ] owned. |
| TACRED | The leader of the group, [ Laura Silsby ], a businesswoman who describes herself as a missionary as well, has also come under scrutiny at home in [ Idaho ], where employees complain of unpaid wages and the state has placed liens on her company bank account. The relationship between [ Laura Silsby ] and [ Idaho ] is | relationship between [ Laura Silsby ] and [ Idaho ] = *state or provinces of residence* |
| FewRel | In June 2017 President of Catalonia Carles Puigdemont announced that a [ referendum ] on [ Catalan independence ] would be held on 1 October 2017. The relationship between [ referendum ] and [ Catalan independence ] is | relationship between [ referendum ] and [ Catalan independence ] = *main subject* |

| Dataset | Input | Output |
|---|---|---|
| CoNLL2005 (SRL) | Still, one federal appeals court has signaled it's willing to entertain the notion, and the lawyers have renewed their arguments in Texas and eight other states where the defense is [ permitted ] under state law. | Still, one federal appeals court has signaled it's willing to entertain the notion, and the lawyers have renewed their arguments in [ Texas and eight other states \| *AM-LOC* ] [ where \| *R-AM-LOC* ] [ the defense \| *A1* ] is permitted [ under state law \| *AM-LOC* ]. |
| CoNLL2012 (SRL) | I'm just as delighted now, nine years later, to be able to welcome you here and to learn about the great changes which have [ occurred ] in your country since I was there. | I'm just as delighted now, nine years later, to be able to welcome you here and to learn about [ the great changes \| *ARG1* ] [ which \| *R-ARG1* ] have occurred [ in your country \| *location* ] [ since I was there \| *ARGM-TMP* ]. |
| ACE2005 (event trigger id.) | Hoon said Saddam's regime was crumbling under the pressure of a huge air assault. | Hoon said Saddam 's regime was [ crumbling \| *end organization* ] under the pressure of a huge air [ assault \| *attack* ]. |
| ACE2005 (event argument cl.) | Chairman Jack Welch is seeking work-related documents of his estranged wife in his high-stakes [ divorce \| *divorce* ] case. | Chairman Jack Welch is seeking work-related documents of [ his estranged wife \| *individual* \| *person* = divorce ] in [ his \| *individual* \| *person* = divorce ] high-stakes divorce case. |
| CoNLL2012 (coreference res.) | What's your new TV series coming up? Oh it's a new show for FX called Beef. Beef? Beef. You start shooting? Um I'm going to Shriport tomorrow. Shriport Louisiana. | What's [ [ your ] new TV series coming up ]? Oh [ it \| *your new TV series coming up* ]'s a new show for FX called Beef. [ Beef? \| *it* ] [ Beef. \| *Beef?* ] [ You \| *your* ] start shooting? Um [ I \| *You* ]'m going to [ Shriport ] tomorrow. [ Shriport Louisiana. \| *Shriport* ] |
| MultiWOZ | [ user ] **:** am looking for a place to to stay that has cheap price range it should be in a type of hotel [ agent ] **:** okay, do you have a specific area you want to stay in? [ user ] **:** no, i just need to make sure it s cheap. oh, and i need parking | [ belief ] *hotel area* not given, *hotel book day* not given, *hotel book people* not given, *hotel book stay* not given, *hotel internet* not given, *hotel name* not given, *hotel parking* yes, *hotel price range* cheap, *hotel stars* not given, *hotel type* hotel [ belief ] |

