# OpenReview forum: "Structured Prediction as Translation between Augmented Natural Languages"
_ICLR.cc/2021/Conference — ICLR 2021 Spotlight_

### Official Review · AnonReviewer4 · 2020-10-25
**AnonReviewer4**

**Rating:** 7
**Confidence:** 4

**Review:**

This paper proposed TANL, a novel approach by using generative models to solve structured prediction tasks in NLP. The key idea is that we can formulate this as a translation from natural language input to the augmented natural language with the structure of the input, and we can leverage the label semantics of the label in the augmented natural language output. In this way, it enables transfer learning from large pretrained generative language models such as T5. This augmented natural language unifies the input/output format for many structured prediction tasks in NLP and thus can facilitate multi-task learning. The experiments cover a dozen datasets on seven different structured prediction tasks (CoNLL04, ADE, NYT, ACE2005 for Entity Relation Extraction, CoNLL03, OntoNotes, GENIA, ACE2005 for NER, TACRED, FewRel 1.0 for Relation Classification, CoNLL05 WSJ, CoNLL05 Brown, CoNLL2012 for SRL, ACE2005 for Event Extraction, CoNLL2012 for Coreference Resolution, MultiWOZ for DST). The result demonstrates that the proposed approach can achieve SOTA on several of them for Entity Relation Extraction, Relation Classification, SRL.

Strengths:
- The idea of using the generative approach to solve structured prediction tasks in NLP is very novel to me, and the author shows there is a possible output format unification for a wide range of tasks and datasets, and thus we can study them all together using a single model even with the same set of hyperparameters.

- The format of output augmented natural language is simple while being able to encode different structures even with nested entities.

- The experimental results show that this generative approach even has superior performance while being much simpler than other task-specific classification models which require careful model architecture design for different tasks.

- The author also shows that the proposed approach is data-efficient and has an advantage in low-resource settings.

Weakness:
- The experiments show that the proposed approach does not perform very well on two tasks: CoNLL-2012 for coreference resolution and MultiWOZ 2.1 for Dialog State Tracking. For CoNLL-2012, TANL has Avg.F1 72.8 while the SOTA CorefQA has 79.9. The author mentioned that CorefQA uses additional question answering pretraining, but TANL is based on T5 which also uses large-scale pretraining on question answering, translation, and summarization. On MultiWOZ 2.1, TANL has 50.5 while the SOTA has 55.7

- The advantage of multi-dataset or multi-task learning is not obvious. The only major improvement from this is on CoNLL04, ADE, CoNLL05 Brown, possibly due to the fact that they are small datasets.

- There is no significance test for the results in Table 1.

Questions:
- Have you tried with other T5 models such as T5-large? or other generative models such as BART/GPT-2?
- I didn't understand your explanation for why the multi-task model has lower scores on coreference resolution. Do you mean you only use smaller training data?
- Why there is no multi-dataset for Relation Classification?
- Why there is no multi-task or multi-dataset for DST?
- How do you compare your methods with Athiwaratkun et al., 2020

---

> ### Author Response · Authors · 2020-11-22
> **Response**
>
> Thank you for your comments.
>
> >The experiments show that the proposed approach does not perform very well on two tasks: CoNLL-2012 for coreference resolution and MultiWOZ 2.1 for Dialog State Tracking. For CoNLL-2012, TANL has Avg.F1 72.8 while the SOTA CorefQA has 79.9. The author mentioned that CorefQA uses additional question answering pretraining, but TANL is based on T5 which also uses large-scale pretraining on question answering, translation, and summarization. On MultiWOZ 2.1, TANL has 50.5 while the SOTA has 55.7
>
> We changed our comment on CorefQA, since (as you correctly pointed out) T5 also is pre-trained on many different tasks. While TANL is less effective than CorefQA, it performs comparably to the next best models in the literature.
> Regarding dialogue state tracking, it is possible that there is a more suitable input/output format that can perform better for this challenging task. Please see our discussion with AnonReviewer2 for more details.
>
> >The advantage of multi-dataset or multi-task learning is not obvious. The only major improvement from this is on CoNLL04, ADE, CoNLL05 Brown, possibly due to the fact that they are small datasets.
>
> We include multi-task / multi-dataset results not only to explore potential gains in performance (which are more apparent for small datasets, as you noted), but also to highlight TANL's ability to train on a variety of tasks without modification to the network architecture. To the best of our knowledge, our multi-task model is the first to handle a wide variety of tasks/datasets simultaneously under a single model, without any additional task-specific prediction modules or parameters. This streamlined approach is a step closer towards building a unified NLP framework and can practically simplify model deployment in real-world applications.
>
> >There is no significance test for the results in Table 1.
>
> We added a new table (Table 2) in the appendix, reporting the standard deviation for all datasets of joint entity-relation extraction and NER.
>
> >Have you tried with other T5 models such as T5-large? or other generative models such as BART/GPT-2?
>
> (This answer is the same as one given to AnonReviewer3)
> We carried out experiments with T5-large, BART-base, and BART-large. In these early experiments, T5-large has shown comparable performance with T5-base, whereas both BART models have shown a drop in performance (around 80 entity F1 and 51 relation F1 for CoNLL04, compared to 89 and 71 obtained with T5). However, we do not have enough evidence that these are the best results obtainable by BART. We decided not to include these results, as they would not shed light on our core contribution. We added a paragraph in the Experiments section about investigating other models.
>
> >I didn't understand your explanation for why the multi-task model has lower scores on coreference resolution. Do you mean you only use smaller training data?
>
> We clarified this point in Section 5.1. For the multi-task experiment we use the same maximum sequence length of 512 across all tasks. The maximum length of 512 is suitable for most datasets but CoNLL-2012 coreference dataset typically requires longer context for optimal performance due to large document size.
>
> >Why there is no multi-dataset for Relation Classification?
>
> We do not perform multi-dataset and multi-task training on FewRel since FewRel is a few-shot benchmark meant to be fine-tuned only on limited data.
>
> >Why there is no multi-task or multi-dataset for DST?
>
> We re-ran our multi-task experiment, to include DST and event extraction. The updated results are reported in Table 1.
>
> >How do you compare your methods with Athiwaratkun et al., 2020
>
> For flat NER, our approach is similar to Athiwaratkun et al., 2020, as mentioned in appendix A.2. The main difference is that we add an alignment step when decoding the output sentence. The results (in Table 1) are comparable: we perform better on CoNLL03, whereas we perform slightly worse on the larger Ontonotes dataset.

---

### Official Review · AnonReviewer2 · 2020-10-28
**Straightforward application of T5 with some strong results**

**Rating:** 6
**Confidence:** 4

**Review:**

This paper presents a text-to-text translation approach to a variety of structured prediction problems. The authors explore several ways to represent each structured prediction problem as a text-to-text translation task and finetune T5 (Raffel et al., 2020) to perform each task. The resulting model gives better results than existing models in the tasks of joint entity relation extraction, relation classification, and semantics role labeling.

Although this work is a rather straight-forward application of T5 to structured prediction problems, the reported experimental results and lessons learned regarding good text-to-text representations on the extensive set of structured prediction tasks should be useful to the community. The experimental results in the multi-dataset and multi-task settings are also interesting (although not much analysis is given in the paper).

I was wondering why the authors did not apply their approach to the task of syntactic parsing, which is probably the most well-studied structured prediction task in NLP. Is there any difficulty in applying the same technique to dependency or phrase-structure parsing?

I think the authors should also discuss the limitations of their approach. Are there any structured prediction tasks in NLP that are difficult to tackle with their approach?

In section 5.2, it is a bit surprising that the model was able to learn the correct output format with only 9 sentences. Were there no invalid outputs?

Minor comments:
p.3: the current state-of-the-art -> the current state of the art?
p.4: dynamic-programming -> dynamic programming (DP)?
p.5: don’t -> do not?
p.5: previous state-of-the-art -> previous state of the art?

---

> ### Author Response · Authors · 2020-11-22
> **Response**
>
> Thank you for your review.
>
> > I was wondering why the authors did not apply their approach to the task of syntactic parsing, which is probably the most well-studied structured prediction task in NLP. Is there any difficulty in applying the same technique to dependency or phrase-structure parsing?
>
> Thank you for the insightful comment regarding syntactic / dependency parsing (DP). The main reason why we did not apply our approach to dependency parsing is that we wanted to focus on semantic rich tasks where we could better demonstrate the model’s ability to exploit semantic knowledge about the labels from pre-trained language models. We leave DP as a future work and give here a suggested augmented natural language format for this task.
>
> Given the following DP tree:
>
> ```
>      root
>       |
>       | +-------dobj---------+
>       | |                    |
> nsubj | |   +------det-----+ | +-----nmod------+
> +--+  | |   |              | | |               |
> |  |  | |   |      +-nmod-+| | |      +-case-+ |
> +  |  + |   +      +      || + |      +      | |
> I  prefer the  morning   flight through  Denver
> ```
>
> The correspondent sentence in augmented natural language is:
>
> *[I | noun subject | prefer ] [prefer | root | prefer ]  [ the | determiner | flight ]  [ morning | noun modifier | flight ]   [ flight | direct object | prefer ]  [ through | case | Denver ]  [ Denver | noun modifier | flight ]*
>
> Where each word word of the original sentence is translated to the format [ word | role | head ].
>
> >I think the authors should also discuss the limitations of their approach. Are there any structured prediction tasks in NLP that are difficult to tackle with their approach?
>
> We found coreference resolution to be challenging due to the necessity to split documents into chunks, which makes it more difficult for our end-to-end model to connect mentions across different chunks. This particularly hurts the performance of our multi-task model, which we trained with a maximum sequence length that is appropriate to most datasets but too small for the documents of the coreference dataset. We expanded on this in Section 5.1.
>
> In addition, we found that the choice of input/output format is important in our framework. Most experiments worked well with the first format choice we tried. However, there are more challenging tasks such as dialogue state tracking which can benefit from more extensive format search. While this might seem ad-hoc, designing input and output format in our framework is akin to designing classification modules and prediction spaces for a discriminative approach, with the difference being that in the end, all tasks in our framework share a unified text-to-text structure and can be trained together easily.
>
> >In section 5.2, it is a bit surprising that the model was able to learn the correct output format with only 9 sentences. Were there no invalid outputs?
>
> We added an Appendix C with a detailed analysis of different kinds of errors in the various data regimes. With only 9 training sentences, about 21% of the output sentences have incomplete patterns (e.g., not all "[" have a corresponding "]" or a "|" is missing). These patterns cannot be decoded into structured objects and count as incorrect predictions. Note that even with these mistakes in the low-resource regimes, our model can still outperform other approaches (+11 entity F1 and +8 relation F1 over SpERT).
>
> >Minor comments: p.3: the current state-of-the-art -> the current state of the art? p.4: dynamic-programming -> dynamic programming (DP)? p.5: don’t -> do not? p.5: previous state-of-the-art -> previous state of the art?
>
> Thank you, we made all suggested corrections.

---

### Official Review · AnonReviewer1 · 2020-10-29
**Good paper, a few suggestions**

**Rating:** 8
**Confidence:** 4

**Review:**

# Summary
In this paper, the authors proposed a unified seq2seq model for structured prediction tasks in NLP. They let the seq2seq model produce mixed outputs of special tokens and the original sentence. Different NLP tasks, including relation classification, entity relation extraction, NER, etc. can be converted into this seq2seq problem by adding special tokens.  The experiments show that the proposed model does better than the previous state-of-the-art, albeit with the help of multi-task and multi-dataset learning, on some of the tasks/datasets.

## Pros:
1. A unified framework that allows for multi-task and multi-dataset learning. Their experiments also show that their model could benefit from multi-task, multi-dataset learning. The experiments on few-shot relation extraction show that their model could transfer knowledge from high-resource tasks to low-resource tasks.
2. The formulation is neat and extensible. More difficult structured predictions tasks (in terms of structure), e.g. dependency parsing, are in principle convertible to this format, although the authors didn't try it on parsing.

## Questions:
For structured prediction tasks, searching for the best output is a crucial part. However, this paper doesn't explore search strategies too much. Only in the appendix, beam search is mentioned. More concretely, I would suggest the authors try to answer the following questions:
1. How much could we improve the current model purely by using better the searching strategy (the headroom)? Different from CRF, the structured prediction model is not markovian, which means we have to brute-force the best output. Is it possible to calculate such an upper-bound performance of the current model? If so, what would be the upper-bound for each task?
2. In this paper, the DP alignment method is a post hoc method. What if we add such a monotonic alignment to the decoding process?

To summarize, I think this is a good paper in terms of extensive experiments and convincing results, but the search strategy still needs to be explored and justified for structured prediction tasks.

---

> ### Author Response · Authors · 2020-11-22
> **Response**
>
> Thank you for your review.
>
> > How much could we improve the current model purely by using better the searching strategy (the headroom)? Different from CRF, the structured prediction model is not markovian, which means we have to brute-force the best output. Is it possible to calculate such an upper-bound performance of the current model? If so, what would be the upper-bound for each task?
>
> We used beam search in our experiments, but we have found that it gives very little advantage over greedy decoding, even when using a larger number of beams (e.g., 32). We clarified this in Appendix A.
> Due to the combinatorially large search space, the actual upper bound performance is infeasible to compute. However, if we accept the beam search as a loose upper bound, there is not much headroom for performance gain through searching.
>
> > In this paper, the DP alignment method is a post hoc method. What if we add such a monotonic alignment to the decoding process?
>
> This is a very good point. We considered it, but in all our experiments the model is precise enough in outputting the right structure (as shown in the newly added Appendix C) that we would not obtain a significant increase in performance. However, on tasks with a more complex structure (e.g., heavily nested) it may bring a larger improvement. Future work using our method on tasks with more complex structure should definitely consider this idea.

---

### Official Review · AnonReviewer3 · 2020-10-29
**A good and novel idea on structured prediction.**

**Rating:** 6
**Confidence:** 4

**Review:**

Recently, multiple research papers focus on task transformations by bridging the gap between different tasks[1,2,3,4]. The original idea may go back to [5]. This paper follows this line of research ideas by reducing a structured prediction problem to a translation problem. The general idea is novel and very interesting. By defining several manually-designed rules, multiple structured outputs are transformed into the output of the translation model. The writing is clear and well-structured.

Pros:
 - A novel and interesting idea for formulating structured prediction tasks to translation problems. This idea is well-motivated in low-resource scenarios and multi-task learning settings.
 - The general framework is easy to implement (only requiring some scripts).



Cons:
 - My main concern with the proposed approach is the decoding process. If the translated sequence has a nice structure, the transformation process is well performed. However, the translated results might be invalid for a specific task. For example, in CoNLL NER, a nested or overlapping structure might be generated. It may need specially designed rules to filter them out. However, this paper does not have many discussions on this point. I would like to know more about this part.
 - I also would like to know the effectiveness of different pre-trained language models. In this paper, a T5-base model is utilized. It might be beneficial to know the empirical effectiveness of different kinds of language models.
 - Some words are not precise. For example, the phrase "generative models" are frequently used to illustrate the translation model. However, in the ML field, generative models may indicate the models that have a generative process of data and model the joint distribution of observed samples.


I am willing to increase my score if some of the questions are well clarified by the authors.

[1] Strzyz et al. Viable Dependency Parsing as Sequence Labeling, NAACL 2019

[2] Yu et al. Named entity recognition as dependency parsing, ACL 2020

[3] Gómez-Rodríguez et al. Constituency parsing as sequence labeling, EMNLP 2018

[4] Li et al. A Unified MRC Framework for Named Entity Recognition. ACL 2020

[5] Vinyals et al. Grammar as a Foreign Language.

---

> ### Author Response · Authors · 2020-11-22
> **Response**
>
> Thank you very much for your comments.
>
> >My main concern with the proposed approach is the decoding process. If the translated sequence has a nice structure, the transformation process is well performed. However, the translated results might be invalid for a specific task. For example, in CoNLL NER, a nested or overlapping structure might be generated. It may need specially designed rules to filter them out. However, this paper does not have many discussions on this point. I would like to know more about this part.
>
> If the model generates a nested pattern where there shouldn't be (as is the case for CoNLL NER), the decoding process can still parse the output to structured objects. However, these objects can disagree with the ground truth labels during evaluation, and will be counted as incorrect. If the output is "invalid" (cannot be parsed into objects) which can be due to incomplete bracket patterns [ ... | ... ], then parts of the sentence are dropped. We expanded the explanation of the decoding process in Section 3 to clarify this point.
>
> To give readers a better understanding of the types of generation errors our model can make, we added a detailed study in Appendix C. For instance, on the CoNLL04 dataset (which is the smallest one we use), about 98.9% of the test output sentences have complete patterns. This number decreases as we reduce the size of the training dataset, down to 78.5% when we only use 9 training sentences. Despite the lowered % of complete patterns, we are able to outperform a discriminative BERT-based model (+11 entity F1 and +8 relation F1) due to our model's data efficiency.
>
> Other generation errors in our analysis includes label type errors, entity errors, or input replication errors. We found that replication errors are the most common ones (82.0% of the sentences are replicated exactly when training on the entire CoNLL04 dataset), whereas other kinds of errors only appear when using a small portion of the training data.
>
> >I also would like to know the effectiveness of different pre-trained language models. In this paper, a T5-base model is utilized. It might be beneficial to know the empirical effectiveness of different kinds of language models.
>
> We carried out experiments with T5-large, BART-base, and BART-large. In these early experiments, T5-large has shown comparable performance with T5-base, whereas both BART models have shown a drop in performance (around 80 entity F1 and 51 relation F1 for CoNLL04, compared to 89 and 71 obtained with T5). However, we do not have enough evidence that these are the best results obtainable by BART. We decided not to include these results, as they would not shed light on our core contribution. We added a paragraph in the Experiments section about investigating other models.
>
> >Some words are not precise. For example, the phrase "generative models" are frequently used to illustrate the translation model. However, in the ML field, generative models may indicate the models that have a generative process of data and model the joint distribution of observed samples.
>
> We acknowledge that the term "generative" is overloaded and can cause confusion. We could alternatively say "sequence to sequence" or "conditional generation".
> The term "generative" emphasizes that our model performs prediction via sequence generation and is flexible to generate many possible output sequences which can differ in terms of patterns depending on the task. This aspect is quite different from the encoder-based methods with additional classification modules where the output structure is rigid -- different architectures are often required for different tasks, if the tasks are not similar enough.

---

### Decision · Program_Chairs · 2021-01-07
**Final Decision**

**Decision:**

Accept (Spotlight)

**Comment:**

This paper proposes a general framework to use MT to solve structural prediction problems.
The method is well developed and be verified in an arrange of tasks including entity recognition, relation classification, event extraction, semantic role labelling, coreference resolution and dialog state tracking and achieves new state-of-the-art in some of these tasks.
Further experiments also suggest the method is especially effect for low resource scenario, if the label semantics can be used appropriately.Further experiments also suggest the method is especially effect for low resource scenario, if the label semantics can be used appropriately.
All reviewers agreed to accept the paper and gave very positive comments.  Some reviewers pointed out that the methods do not improve the performance significantly in some of the tasks.  And more analysis is wanted (by reviewer1).